# FUSE: Frequency-domain Unification and Spectral Energy Alignment for Multi-modal Object Re-Identification

Xuanhao Qi[1]  Tom H. Luan[1]  Yukang Zhang[2]  Jinkai Zheng[1]  Zhou Su[1]  Shuwei Li[3]  Lei Tan[3]

## Abstract

Despite significant progress in multi-modal Re-Identification (ReID), existing methods tend to emphasize low-frequency cues. Consequently, they focus on attributes such as color, illumination, and coarse appearance, while overlooking mid and high-frequency structures that encode geometric, textural, and identity-discriminative details. This imbalance leads to incomplete spectral representations and unstable cross-modal alignment. To overcome these limitations, we introduce FUSE, a frequency-domain framework that reformulates multi-modal ReID as a two-stage process of spectral disentanglement and energy alignment. The proposed Spectral Decomposition Module (SDM) adaptively partitions features into low, mid, and high-frequency subspaces, enabling hierarchical spectral modeling. The Cross-Modal Alignment Module (CAM) further enforces energy alignment and subspace complementarity across modalities via frequency-consistency regularization. In addition, FUSE incorporates learnable frequency modulation to enhance robustness under varying illumination and heterogeneous sensor conditions. Extensive experiments on RGBNT201, RGBNT100, and MSVR310 show that FUSE achieves 9.1% mAP and 9.5% Rank-1 improvements, establishing an interpretable frequency-domain paradigm for multi-modal representation learning.

## 1. Introduction

Multi-modal re-identification (ReID) aims to match the same individual across non-overlapping cameras by inte-

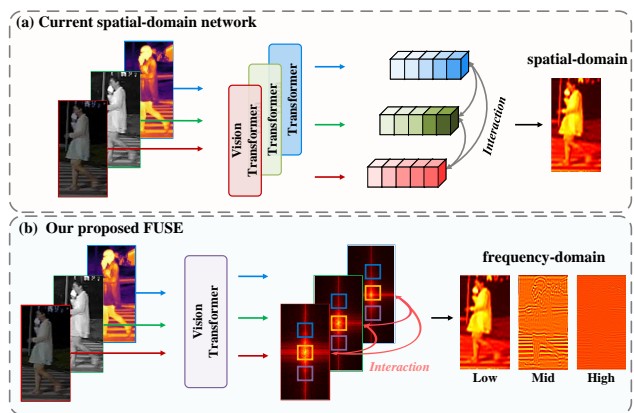

*Figure 1.* **Comparison between the proposed FUSE and mainstream spatial-domain structures.** (a) Existing multi-modal ReID methods mainly rely on spatial domain fusion, but the inherent low-frequency bias (Park & Kim, 2022; Wang et al., 2020) of both CNNs and ViTs causes models to predominantly capture global low-frequency semantics while neglecting mid and high-frequency details, leading to incomplete spectral representation and unstable cross-modal alignment. (b) The proposed FUSE explicitly models in the frequency domain, where spectral decomposition and energy alignment partition features into low, mid, and high-frequency subspaces, achieving semantically interpretable cross-modal feature fusion.

grating visual information from RGB, near-infrared (NIR), and thermal infrared (TIR) modalities. This capability is essential for reliable perception in intelligent surveillance, autonomous driving, and cross-environment visual understanding. While deep learning–based single-modal ReID has achieved notable success (He et al., 2021; Luo et al., 2019; Sun et al., 2018), its robustness remains limited under complex environmental variations (Ye et al., 2021; Niu et al., 2026; Yang et al., 2022). Multi-modal ReID improves robustness by integrating complementary information across modalities, yet cross-modal feature alignment remains fragile under realistic perception conditions.

Recent research in multi-modal ReID has focused on identity-level fusion (Wang et al., 2024a; 2025b;a), fine-grained feature complementarity (Feng et al.), and frequency-domain interaction modeling (Yang et al., 2025; Zhang et al., 2024), achieving considerable progress. However, most existing approaches remain limited to spatial or semantic alignment and fail to capture spectral-level cross-modal discrepancies. As illustrated in Figure 1(a), existing

[1]School of Cyber Science and Engineering, Xi'an Jiaotong University, Xi'an, China [2]School of Informatics, Xiamen University, Xiamen, China [3]National University of Singapore, Singapore. Correspondence to: Tom H. Luan <tom.luan@xjtu.edu.cn>, Shuwei Li <Shuwei@u.nus.edu>.

*Proceedings of the 43rd International Conference on Machine Learning*, Seoul, South Korea. PMLR 306, 2026. Copyright 2026 by the author(s).

multi-modal ReID methods rely on CNN- and ViT-base (Ma et al., 2024) spatial feature extraction and fusion (Ma et al., 2026; 2023b), yet these architectures inherently favor low-frequency responses (Park & Kim, 2022; Wang et al., 2020). The resulting low-frequency dominance limits the ability to capture cross-modal mid and high-frequency differences and obscures intrinsic spectral disparities across modalities, resulting in unstable cross-modal alignment, where mid and high-frequency bands carry structurally and semantically discriminative information (Zhang et al., 2025b). As a result, the frequency representation becomes incomplete and ultimately leads to unstable cross-modal alignment (Bo et al., 2025; Ma et al., 2023a). These observations highlight the limitations of spatial-domain modeling and motivate a reformulation of cross-modal fusion from a frequency-domain perspective, where spectral structures can be explicitly disentangled and coordinated.

To address the above issue, we propose **FUSE**, a unified frequency-domain framework for multi-modal re-identification. FUSE formulates the task as a two-stage process of spectral disentanglement and energy alignment, enabling robust and interpretable representation unification in the frequency space. In the first stage, the Spectral Decomposition Module (SDM) adaptively decomposes features into low, mid, and high-frequency subspaces using learnable soft bandpass filters and applies energy-balance regularization to preserve mid-frequency dominance for stable structural consistency. Moreover, SDM adopts differentiated convolutional strategies across frequency bands, where dilated, standard, and depthwise separable convolutions are applied to capture global semantics, maintain structural coherence, and extract fine-grained textures, respectively. In the second stage, the Cross-Modal Alignment Module (CAM) computes band-wise amplitude statistics and learns affine parameters to align spectral energy and maintain subspace complementarity. During training, FUSE introduces adaptive spectral masks for dynamic perturbation and masking, improving robustness under illumination imbalance and modality noise. This two-stage design transforms cross-modal fusion from implicit spatial correlation into explicit spectral reasoning, achieving frequency-based, semantically consistent, and interpretable feature alignment.

The contributions of this paper are summarized as follows:

- We propose a frequency-domain framework named FUSE for multi-modal ReID. FUSE explicitly models inter-band interactions in the frequency domain, enabling more complete spectral representations and enhancing multi-spectral object ReID.

- We introduce a Spectral Decomposition Module (SDM) and a Cross-Modal Alignment Module (CAM), where SDM separates features into low, mid, and high-

frequency subspaces to capture complementary spectral semantics, and CAM aligns the spectral energy distribution across modalities to stabilize cross-modal feature learning.

- Extensive experiments conducted on three public multi-modal ReID datasets, namely RGBNT201, RGBNT100, and MSVR310, demonstrate the superior performance of FUSE, improvement a 9.1% mAP and 9.5% Rank-1 on RGBNT201.

## 2. Related Work

Multi-modal object ReID improves recognition robustness by leveraging heterogeneous spectral inputs such as RGB, NIR, and TIR. Most prior work designs spatial-domain fusion mechanisms. Early CNN-based methods include PFNet (Zheng et al., 2021), which adopts progressive multi-spectral fusion for low-illumination robustness, and IEEE (Wang et al., 2022), which introduces information exchange and intra-class discrepancy maximization to strengthen modality-specific representations. To exploit structural relations across modalities, GPFNet (He et al., 2023) proposes graph-guided progressive fusion that aggregates multi-modal features by reasoning over relational topology. With the rise of ViTs, recent approaches explore token-level interaction. H-ViT (Pan et al., 2022) introduces modality tokens at multiple stages to balance shared and private representations, while TOP-ReID (Wang et al., 2024a) uses cyclic token permutation for fine-grained alignment. MambaPro (Wang et al., 2025a) leverages selective state-space aggregation for long-range interaction with linear complexity, improving scalability and robustness. DeMo (Wang et al., 2025b) incorporates modality-aware deformable aggregation to enable flexible token correspondence and alleviate geometric misalignment, and IDEA (Wang et al., 2025c) employs text-inverted semantic guidance with deformable token fusion to enhance shared semantics and reduce multi-spectral gaps. Despite these advances, spatial fusion remains dominant and often struggles under RGB NIR TIR spectral mismatch, biasing networks toward low-frequency cues and under-representing mid and high-frequency identity structure. This motivates frequency-domain modeling to capture spectral heterogeneity and improve cross-modal consistency. FDNM (Zhang et al., 2025b) decomposes features into Fourier amplitude and phase, where amplitude reflects modality variation and phase preserves geometric stability. MFENet (Gu et al., 2025) learns a low high frequency split, where low frequencies encode style and illumination and high frequencies capture identity structure. TIENet (Yang et al., 2025) enhances geometric stability through progressive multi-level spectral interaction. EDITOR (Zhang et al., 2024) uses wavelet decomposition to obtain stable multi-scale frequency features and enables learnable spatial fre-

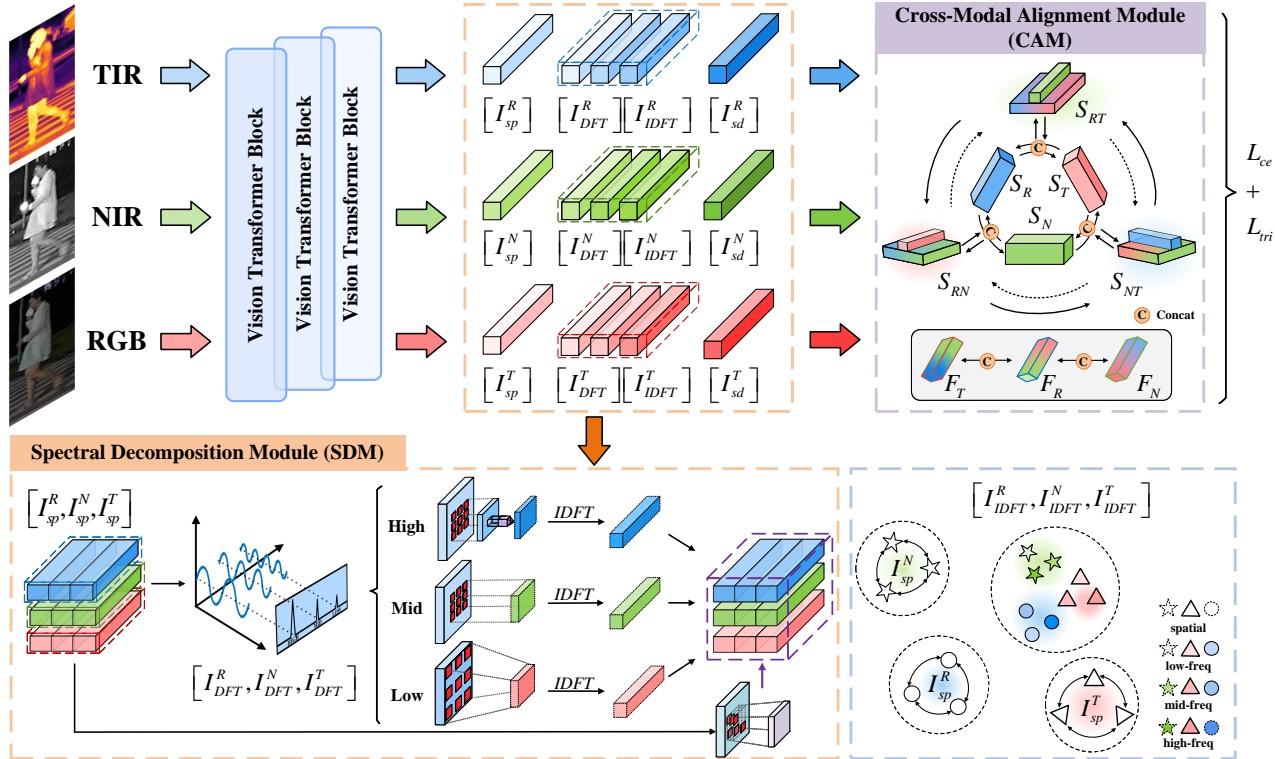

*Figure 2.* **Overall architecture of FUSE.** FUSE leverages frequency-domain modeling to enhance multi-modal person re-identification. Input images from RGB, NIR, and TIR modalities are processed by a shared Vision Transformer backbone to extract spatial features. The Spectral Decomposition Module (SDM) adaptively partitions features into frequency sub-bands and applies specialized enhancement, while the Cross-Modal Alignment Module (CAM) performs frequency-aware interaction and consistency alignment. Identity classification and triplet losses supervise the final fused representation.

quency token selection. Although effective, these methods rely on pre-defined frequency decompositions, lack principled frequency semantics, and omit cross-modal spectral energy alignment, yielding unstable, less interpretable representations (Lu et al., 2024; 2025; Niu et al., 2025; Zheng et al., 2025b). This motivates FUSE, reframing cross-modal ReID through band-wise spectral disentanglement and energy alignment for robust multi-frequency representations.

## 3. Methodology

An overview of the full architecture is presented in Figure 2. We propose FUSE, a modular framework that integrates frequency-domain modeling into multi-modal person re-identification. While existing methods predominantly rely on spatial-domain feature fusion, FUSE decomposes intermediate representations into frequency sub-bands and aligns spectral statistics across modalities. The architecture has two components. The Spectral Decomposition Module (SDM) adaptively partitions features into low, mid, and high-frequency bands and performs band-specific enhancement. The Cross-Modal Alignment Module (CAM) models inter-modal interactions through frequency-aware attention and enforces global spectral consistency via regularization.

By introducing frequency as an axis of representation and alignment, FUSE provides a principled and interpretable approach for reducing cross-modal discrepancies.

### 3.1. Spectral Decomposition Module (SDM)

We introduce SDM from the perspective of spectral discrepancies in Figure 2. RGB, NIR, and TIR exhibit distinct energy patterns across low, mid, and high frequencies, leading to feature mixing and unstable cross-modal alignment. SDM separates and models structural cues in different frequency bands by applying Discrete Fourier Transform (DFT) to ViT features, generating learnable radial masks, and performing band-specific convolutional enhancement to obtain complementary and stable spectral components. For efficient multi-modal representation learning, FUSE adopts a unified ViT backbone shared by RGB, NIR, and TIR, reducing model complexity and promoting early alignment. The tokenized features for each modality are given by:

$$I_{sp}^i = \text{ViT}(I^i), \quad i \in \{R, N, T\}, \tag{1}$$

where $I^R \in \mathbb{R}^{C \times H \times W}$, $I^N \in \mathbb{R}^{C \times H \times W}$, and $I^T \in \mathbb{R}^{C \times H \times W}$ denote the RGB, NIR and TIR images, respectively. The tokenized features $I_{sp}^R$, $I_{sp}^N$ and $I_{sp}^T$, each with

shape $\mathbb{R}^{C \times H \times W}$, are extracted from the last ViT layer.

To uncover spectral characteristics inherent in spatial representations, SDM transforms tokenized RGB, NIR, and TIR features into the frequency domain using the two-dimensional DFT. This operation yields modality-specific Fourier spectra $I_{DFT}^{R}$, $I_{DFT}^{N}$, and $I_{DFT}^{T}$, which provide a unified frequency-domain representation across modalities and serve as input to SDM's learnable radial decomposition.

$$\mathcal{F}(u,v) = \frac{1}{\sqrt{HW}} \sum_{h=0}^{H-1} \sum_{w=0}^{W-1} x(h,w) e^{-j2\pi\left(\frac{hu}{H} + \frac{wv}{W}\right)}, \quad (2)$$

where $j$ denotes the imaginary unit. $(u, v)$ are the discrete frequency indices along the horizontal and vertical axes of the 2D spectrum. $\mathcal{F}(\cdot)$ denotes the 2D DFT, and $\mathcal{F}^{-1}(\cdot)$ denotes the inverse discrete Fourier transform (IDFT).

$$I_{DFT}^{i} = \mathcal{F}(I_{sp}^{i}), \quad i \in \{R, N, T\}. \quad (3)$$

The inherent spectral differences across modalities make robust multi-modal alignment difficult, and fixed-threshold band definitions often fail under such variability. To address this challenge, SDM obtains a learnable frequency representation. SDM abstracts the 2D frequency plane into a radial function centered at the Direct Current (DC) component $(u_0, v_0)$ and partitions along the radial axis. Here, $(u, v)$ denotes frequency-bin indices on the shifted 2D spectrum, $u \in 0, \ldots, H-1$ and $v \in 0, \ldots, W-1$. Under FFT shift, the DC center is fixed at $(u_0, v_0) = \left(\lfloor \frac{H}{2} \rfloor, \lfloor \frac{W}{2} \rfloor\right)$. The maximal usable radius is $R_{\max} = \frac{1}{2} \min(H, W)$. For any $(u, v)$, the radial distance is computed as:

$$d(u,v) = \sqrt{(u - u_0)^2 + (v - v_0)^2}. \quad (4)$$

The frequency allocation is parameterized by a three-dimensional vector $\boldsymbol{\theta} = [\theta_L, \theta_M, \theta_H]$. This vector is a set of model weights optimized end-to-end via backpropagation. Its Softmax-normalized form defines the learnable, relative proportions $[r_L, r_M, r_H]$ of the allocation:

$$[r_L, r_M, r_H] = \text{softmax}(\boldsymbol{\theta}). \quad (5)$$

These proportions adaptively induce three continuous and non-overlapping radial intervals within $R_{\max}$, corresponding to the Low-Frequency $R_L$, Mid-Frequency $R_M$, and High-Frequency $R_H$ subspaces:

$$\begin{aligned}
R_L &= [0, \, r_L R_{\max}], \\
R_M &= (r_L R_{\max}, \, (r_L + r_M) R_{\max}], \quad (6) \\
R_H &= ((r_L + r_M) R_{\max}, \, R_{\max}].
\end{aligned}$$

Each frequency coordinate is then assigned to one of these intervals according to its distance $d(u, v)$, producing mutually exclusive annular masks $M_i$:

$$M_i(u,v) = \mathbb{I}\big[d(u,v) \in R_i\big], \qquad i \in \{L, M, H\}. \quad (7)$$

where $\mathbb{I}[\cdot]$ denotes the indicator function that equals 1 if the condition holds and 0 otherwise.

To preserve structural advantages from frequency disentanglement, each spectral subspace is processed with an operator whose spatial inductive bias matches its frequency band. Low-frequency features $M_L$ vary slowly across space and encode global semantics, so dilated convolution ($d=2$) enlarges the receptive field to capture smooth patterns. Mid-frequency features $M_M$ represent meso-scale structures such as contours and local shapes, making standard $3 \times 3$ convolution a natural fit. High-frequency features $M_H$ contain fine textures and sharp transitions; depthwise convolution preserves channel-wise details while avoiding over-smoothing, maintaining discriminative high-frequency cues.

$$X_L = \text{DConv}_{3\text{x}3}(M_L), \quad (8)$$

$$X_M = \text{Conv}_{3\text{x}3}(M_M), \quad (9)$$

$$X_H = \text{DWConv}_{3\text{x}3}(M_H). \quad (10)$$

To restore a complete spatial representation from the disentangled components, we apply the Inverse Discrete Fourier Transform (IDFT). The enhanced spatial feature maps $(X_L, X_M, X_H)$ are aggregated by summation, combining their enhanced spectral components. The two-dimensional inverse transform $\mathcal{F}^{-1}$ is then applied to the aggregated result to obtain enhanced spatial features $I_{IDFT}^{i}$:

$$I_{IDFT}^{i} = \mathcal{F}^{-1}(X_H + X_M + X_L), \quad i \in \{R, N, T\}. \quad (11)$$

To capture modality-shared channel dependencies, we first apply Global Average Pooling (GAP) to RGB, NIR, and TIR features to obtain compact channel descriptors. These descriptors are then refined by a lightweight 1D convolution with a kernel size $k$, and a sigmoid $\sigma$ activation produces the normalized channel weights:

$$X_C^{i} = \sigma(\text{Conv}_{1D}^{k}(\text{GAP}(I_{sp}^{i}))), \quad i \in \{R, N, T\}. \quad (12)$$

With the frequency-enhanced features and modality-shared channel weights in place, SDM integrates them into the original spatial representation through residual modulation. The enhanced feature $I_{IDFT}^{i}$ is first reweighted by the channel attention map $X_C^{i}$, and the result is added to the original ViT feature $I_{sp}^{i}$ to form the final decomposed representation:

$$I_{sd}^{i} = I_{sp}^{i} + \big(I_{IDFT}^{i} \odot X_C^{i}\big), \quad i \in \{R, N, T\}. \quad (13)$$

This residual fusion preserves spatial semantics while injecting frequency-aware structural cues, yielding a more stable and discriminative multi-modal representation.

### 3.2. Cross-Modal Alignment Module (CAM)

As illustrated in Figure 3, SDM disentangles features into frequency sub-bands, but RGB, NIR, and TIR still show distinct amplitude and energy patterns that cause cross-modal

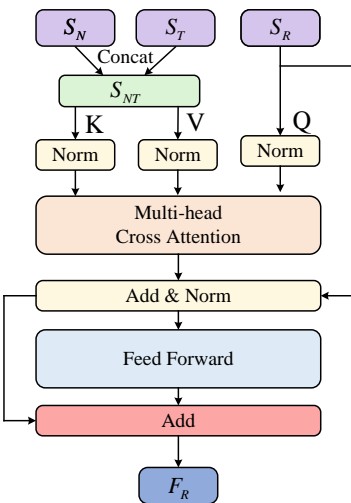

*Figure 3.* **The architecture of the Cross-Modal Alignment Module (CAM).** It utilizes the concatenated NIR and TIR tokens ($S_{NT}$) as Key/Value and uses RGB tokens $S_R$ as the Query for refinement via multi-head cross-attention.

inconsistency. To address this, we design the Cross-Modal Alignment Module (CAM), which leverages frequency-domain complementarity and spectral alignment to reduce modality-specific biases. CAM performs frequency-aware cross-modal attention and introduces an energy-consistency constraint to produce stable fused representations. CAM takes spatial features $I_{ca}^R, I_{ca}^N, I_{ca}^T$ as inputs and outputs the aligned representation $I_{ca}$ through cross-modal interaction. Without loss of generality, we start with the RGB stream. CAM adopts a Multi-Head Cross-Attention (MHCA) module with $N_h$ heads for all-to-one aggregation across modalities. Concretely, the token sequence $S_R \in \mathbb{R}^{M \times D}$ from $I_{ca}^R$ is projected to form the query matrix $Q \in \mathbb{R}^{M \times D}$. The aggregated tokens $S_{NT} = Concat(S_N, S_T) \in \mathbb{R}^{2M \times D}$ from NIR and TIR are projected to produce key $K$ and value $V$. The interaction between $I_{ca}^R$ and modalities $(N, T)$ in the $h$-th head is computed as:

$$\hat{F}_R^h = \sigma(\frac{Q^h K^{h\top}}{\sqrt{d}})V^h, \qquad (14)$$

where $\sigma$ denotes the softmax function and $(\cdot)^\top$ represents matrix transposition. $Q^h \in \mathbb{R}^{M \times d}$, $K^h, V^h \in \mathbb{R}^{2M \times d}$, and $d = D/N_h$. The outputs of $N_h$ heads $(\hat{F}_R^1, \ldots, \hat{F}_R^{N_h})$ are concatenated into $\hat{F}_R \in \mathbb{R}^{M \times D}$.

This operation corresponds to a single-round, all-to-one frequency-aware alignment, where the target modality aggregates complementary cues from the other two modalities. The same symmetric procedure is applied to all modalities:

$$\begin{aligned} F_R &= \text{LN}(\text{MHCA}(S_R, S_{NT})), \\ F_N &= \text{LN}(\text{MHCA}(S_N, S_{RT})), \qquad (15) \\ F_T &= \text{LN}(\text{MHCA}(S_T, S_{RN})). \end{aligned}$$

Aggregating the operations across all modalities, the entire single-round alignment process is succinctly summarized by the following function:

$$F_R, F_N, F_T = \text{CAM}(I_{ca}^R, I_{ca}^N, I_{ca}^T). \qquad (16)$$

Finally, the aligned feature sequences are concatenated to form the fused representation $I_{ca}$:

$$I_{ca} = \text{Concat}(F_R, F_N, F_T). \qquad (17)$$

Unlike TOP-ReID's relay fusion that cyclically permutes class tokens to attend to the next modality's patches, CAM builds a unified K/V by concatenating the other modalities' tokens and applies symmetric all-to-one cross-attention, letting each modality access full multimodal context in one pass for complementary aggregation.

### 3.3. Objective Function

**Frequency Consistency Loss $\mathcal{L}_{freq}$.** To address the spectral discrepancies between heterogeneous modalities and enforce spectral alignment, we introduce the $\mathcal{L}_{freq}$ as a structural regularization term. This loss ensures that modality-specific features, processed by the SDM module, possess consistent global spectral characteristics across modalities. Given the input features $I_{sd}^R, I_{sd}^N, I_{sd}^T$ from each modality, we first apply the 2D DFT to obtain their amplitude spectra $|I_{sd}^R|, |I_{sd}^N|, |I_{sd}^T|$. Then, we extract the compact spectral descriptor $G_R, G_N, G_T$ for each modality by applying GAP across spatial dimensions:

$$G_R, G_N, G_T = \text{GAP}(|I_{sd}^R|, |I_{sd}^N|, |I_{sd}^T|). \qquad (18)$$

The loss function is designed to minimize the average pairwise cosine distance between the global spectral descriptors of different modalities:

$$\mathcal{L}_{freq} = \frac{2}{M(M-1)} \sum_{1 \le i < j \le M} (1 - \frac{G_i G_j}{\|G_i\|_2 \|G_j\|_2}), \quad (19)$$

where $M$ is the number of modalities, and $G_i$ and $G_j$ are frequency descriptors for modalities $i$ and $j$, respectively. Minimizing $\mathcal{L}_{freq}$ encourages feature representations with consistent spectral characteristics across modalities.

The overall training objective is to minimize the total loss function $\mathcal{L}_{total}$, which combines the representation learning objectives with the spectral regularization term. The total loss is defined as:

$$\mathcal{L}_{total} = \mathcal{L}_{id} + \mathcal{L}_{tri} + \lambda_{freq}\mathcal{L}_{freq}, \qquad (20)$$

where $\mathcal{L}_{id}$ denotes the identity loss, $\mathcal{L}_{tri}$ represents the triplet loss, and $\mathcal{L}_{freq}$ is our proposed frequency consistency loss. The hyperparameter $\lambda_{freq}$ balances the strength

*Table 1.* **Performance comparison on RGBNT201.** The best and second results are in bold and underlined, respectively.

| | Methods | RGBNT201 | | | |
|---|---|---|---|---|---|
| | | mAP | R-1 | R-5 | R-10 |
| Single | MUDeep (Qian et al., 2017) | 23.8 | 19.7 | 33.1 | 44.3 |
| | HACNN (Li et al., 2018) | 21.3 | 19.0 | 34.1 | 42.8 |
| | MLFN (Chang et al., 2018) | 26.1 | 24.2 | 35.9 | 44.1 |
| | PCB (Sun et al., 2018) | 32.8 | 28.1 | 37.4 | 46.9 |
| | OSNet (Zhou et al., 2019) | 25.4 | 22.3 | 35.1 | 44.7 |
| | CAL (Rao et al., 2021) | 27.6 | 24.3 | 36.5 | 45.7 |
| Multi | HAMNet (Li et al., 2020) | 27.7 | 26.3 | 41.5 | 51.7 |
| | PFNet (Zheng et al., 2021) | 38.5 | 38.9 | 52.0 | 58.4 |
| | IEEE (Wang et al., 2022) | 47.5 | 44.4 | 57.1 | 63.6 |
| | DENet (Zheng et al., 2023a) | 42.4 | 42.2 | 55.3 | 64.5 |
| | LRMM (Wu et al., 2025) | 52.3 | 53.4 | 64.6 | 73.2 |
| | HTT (Wang et al., 2024b) | 71.1 | 73.4 | 83.1 | 87.3 |
| | EDITOR (Zhang et al., 2024) | 66.5 | 68.3 | 81.1 | 88.2 |
| | RSCNet (Yu et al., 2024) | 68.2 | 72.5 | - | - |
| | TOP-ReID (Wang et al., 2024a) | 72.3 | 76.6 | 84.7 | 89.4 |
| | WTSF-ReID(Yu et al., 2025) | 67.9 | 72.2 | 83.4 | 89.7 |
| | DESANet (Dong et al., 2025) | 74.6 | 77.6 | 87.1 | 91.3 |
| | PromptMA (Zhang et al., 2025a) | 78.4 | 80.9 | 87.0 | 88.9 |
| | DeMo (Wang et al., 2025b) | 79.0 | 82.3 | 88.8 | 92.0 |
| | IDEA (Wang et al., 2025c) | 80.2 | 82.1 | 90.0 | 93.3 |
| | **FUSE** | **81.4** | **86.1** | **91.5** | **93.8** |

*Table 2.* **Performance comparison on RGBNT100 and MSVR310.** The best and second results are in bold and underlined, respectively.

| | Methods | RGBNT100 | | MSVR310 | |
|---|---|---|---|---|---|
| | | mAP | R-1 | mAP | R-1 |
| Single | PCB (Sun et al., 2018) | 57.2 | 83.5 | 23.2 | 42.9 |
| | MGN (Wang et al., 2018) | 58.1 | 83.1 | 26.2 | 44.3 |
| | DMML (Chen et al., 2019) | 58.5 | 82.0 | 19.1 | 31.1 |
| | BoT (Luo et al., 2019) | 78.0 | 95.1 | 23.5 | 38.4 |
| | OSNet (Zhou et al., 2019) | 75.0 | 95.6 | 28.7 | 44.8 |
| | Circle Loss (Sun et al., 2020) | 59.4 | 81.7 | 22.7 | 34.2 |
| | HRCN (Zhao et al., 2021) | 67.1 | 91.8 | 23.4 | 44.2 |
| | TransReID (He et al., 2021) | 75.6 | 92.9 | 18.4 | 29.6 |
| | AGW (Ye et al., 2021) | 73.1 | 92.7 | 28.9 | 46.9 |
| Multi | HAMNet (Li et al., 2020) | 74.5 | 93.3 | 27.1 | 42.3 |
| | PFNet (Zheng et al., 2021) | 68.1 | 94.1 | 23.5 | 37.4 |
| | GAFNet (Guo et al., 2022) | 74.4 | 93.4 | - | - |
| | GPFNet (He et al., 2023) | 75.0 | 94.5 | - | - |
| | CCNet (Zheng et al., 2023b) | 77.2 | 96.3 | 36.4 | 55.2 |
| | HTT (Wang et al., 2024b) | 75.7 | 92.6 | - | - |
| | RSCNet (Yu et al., 2024) | 82.3 | 96.6 | 39.5 | 49.6 |
| | TOP-ReID (Wang et al., 2024a) | 81.2 | 96.4 | 35.9 | 44.6 |
| | EDITOR (Zhang et al., 2024) | 82.1 | 96.4 | 39.0 | 49.3 |
| | LRMM (Wu et al., 2025) | 78.6 | 96.7 | 36.7 | 49.7 |
| | FACENet (Zheng et al., 2025a) | 81.5 | 96.9 | 36.2 | 54.1 |
| | WTSF-ReID (Yu et al., 2025) | 82.2 | 96.5 | 39.2 | 49.1 |
| | MambaPro (Wang et al., 2025a) | 83.9 | 94.7 | 47.0 | 56.5 |
| | DESANet (Dong et al., 2025) | 82.1 | 97.4 | 39.2 | 47.8 |
| | DeMo (Wang et al., 2025b) | 86.2 | **97.6** | 49.2 | 59.8 |
| | IDEA (Wang et al., 2025c) | 87.2 | 96.5 | 47.0 | 62.4 |
| | **FUSE** | **88.5** | 96.9 | **50.1** | **65.7** |

of the spectral regularization. This combined objective ensures that the model learns discriminative feature representations while effectively maintaining cross-modal spectral consistency, thereby significantly enhancing the robustness of feature alignment.

## 4. Experiments

### 4.1. Implementation

**Datasets.** We evaluate our method on three widely used multi-modal ReID datasets. RGBNT201 (Zheng et al., 2021) contains 4,787 aligned RGB, NIR, and TIR images from 201 identities. RGBNT100 (Li et al., 2020) is a large-scale vehicle dataset with 17,250 RGB–NIR–TIR triples collected under diverse and challenging conditions. MSVR310 (Zheng et al., 2022) is a smaller but high-quality vehicle dataset with 2,087 image triples captured across varying environments and time periods.

**Evaluation protocols.** To assess the performance of our method, we adopt mean Average Precision (mAP) and Cumulative Matching Characteristics (CMC) at Rank-1, Rank-5, and Rank-10. These metrics are standard in this field and provide a comprehensive evaluation of the model's effectiveness. Additionally, we report the number of parameters and FLOPs to analyze computational complexity of our model.

**Implementation Details.** We implement our model using

PyTorch and run experiments on an NVIDIA 3090 GPU. The visual encoder is initialized with pre-trained CLIP. Input images are resized to $256 \times 128$ for RGBNT201 and $128 \times 256$ for RGBNT100 and MSVR310. Data augmentation follows standard practice, including random horizontal flipping, cropping, and random erasing. The mini-batch size is set to 128 for RGBNT100 and RGBNT201, and 64 for MSVR310, with dataset-specific sampling strategies. We use Adam with an initial learning rate of $3.5 \times 10^{-4}$, while the encoder learning rate is $5 \times 10^{-6}$. The model is trained for 45 epochs on RGBNT201 and RGBNT100, and 50 epochs on MSVR310.

### 4.2. Comparison with State-of-the-Art Methods[1]

We perform comparisons with state-of-the-art (SOTA) methods on three multi-modal ReID datasets and demonstrate competitive results compared with previous work.

**Multi-modal Person ReID.** In Table 1, we compare our

---

[1]See Sec. A.1 for evaluation under missing-modality scenarios and Sec. A.2 for computational cost and efficiency.

*Table 3.* **Comparison with different modules.** We show the best score in bold.

| Index | Modules | | Metrics | | | | Average |
|-------|---------|-----|------|------|------|------|---------|
| | CAM | SDM | mAP | R-1 | R-5 | R-10 | |
| 1 | × | × | 76.0 | 77.8 | 87.1 | 90.4 | 82.8 |
| 2 | ✓ | × | 77.2 | 80.4 | 87.6 | 91.6 | 84.2 |
| 3 | × | ✓ | 79.7 | 82.1 | 90.2 | 92.3 | 86.1 |
| 4 | ✓ | ✓ | **80.8** | **84.1** | **91.5** | **93.5** | **87.5** |

*Table 4.* **Performance analysis under different frequency for SDM.** We show the best score in bold.

| SDM | | | Metrics | | | | Average |
|-----|-----|------|------|------|------|------|---------|
| Low | Mid | High | mAP | R-1 | R-5 | R-10 | |
| 1 | 0 | 0 | 75.4 | 78.7 | 86.5 | 90.9 | 82.9 |
| 0 | 1 | 0 | 72.2 | 73.4 | 84.1 | 88.4 | 79.5 |
| 0 | 0 | 1 | 69.0 | 73.8 | 83.6 | 88.4 | 78.7 |
| 0.2 | 0.4 | 0.4 | 77.2 | 80.4 | 87.6 | 91.6 | 84.2 |
| 0.4 | 0.2 | 0.2 | 78.5 | 81.2 | 88.2 | 91.9 | 84.9 |
| 0.3 | 0.3 | 0.3 | **79.7** | **82.1** | **90.2** | **92.3** | **86.1** |

*Table 5.* **Comparison with different $\lambda_{freq}$ for FUSE.** We show the best score in bold.

| Scale of $\lambda_{freq}$ | mAP | R-1 | R-5 | R-10 | Average |
|-------|------|------|------|------|---------|
| 0.0 | 80.8 | 84.1 | 91.5 | 93.5 | 87.5 |
| 0.1 | **81.4** | **86.1** | **91.5** | **93.8** | **88.2** |
| 0.2 | 80.2 | 83.1 | 89.8 | 92.7 | 86.5 |
| 0.3 | 79.9 | 82.7 | 90.2 | 92.9 | 86.4 |

*Table 6.* **Comparison with different kernel size for FUSE.** We show the best score in bold.

| Kernel Size | mAP | R-1 | R-5 | R-10 | Average |
|-------|------|------|------|------|---------|
| 3,5,7 | 78.8 | 82.5 | 86.7 | 90.1 | 84.5 |
| 3,7,5 | 81.2 | 84.2 | 91.4 | 93.5 | 87.6 |
| 5,3,7 | 79.4 | 83.0 | 90.4 | 92.2 | 86.3 |
| 5,7,3 | 78.7 | 82.4 | 86.6 | 90.0 | 84.5 |
| 7,3,5 | 81.0 | 84.0 | 91.2 | 93.2 | 87.4 |
| 7,5,3 | 78.6 | 82.3 | 86.5 | 90.0 | 84.5 |
| 3,3,3 | **81.4** | **86.1** | **91.5** | **93.8** | **88.2** |
| 5,5,5 | 80.6 | 83.1 | 90.9 | 93.2 | 87.0 |
| 7,7,7 | 79.3 | 82.8 | 91.1 | 94.4 | 86.9 |

proposed FUSE method with existing single-spectral and multi-modal approaches on the challenging RGBNT201 dataset. The results show that multi-modal methods generally outperformed their single-spectral counterparts, underscoring the effectiveness of leveraging complementary information from different image spectra for multi-modal Person ReID. Among single-spectral methods, PCB achieved the highest performance, with an mAP of 32.8% and an R-1 of 28.1%. Among multi-modal methods, our FUSE model established a new SOTA across all ranking metrics, achieving 81.4% mAP and 86.1% R-1. Compared to the previously prominent baseline TOP-ReID (72.3% mAP and 76.6% R-1), FUSE demonstrated a substantial performance leap, with absolute gains of 9.1% in mAP and 9.5% in R-1. Furthermore, against the next-best multi-modal method, IDEA (80.2% mAP and 82.1% R-1), FUSE showed a further improvement of 1.2% in mAP and a significant 4.0 percentage point increase in R-1. These gains substantiate the efficacy of the FUSE design, which leverages Frequency-domain Unification and Spectral Energy Alignment for robust cross-modal discriminative feature learning.

**Multi-modal Vehicle ReID.** In Table 2, we compare our proposed FUSE method with existing single-spectral and multi-modal approaches on the RGBNT100 and MSVR310 datasets. The results consistently demonstrate that multi-modal methods achieve superior performance compared to single-spectral methods, underscoring the effectiveness of utilizing complementary cross-modal information. Among single-spectral methods, OSNet and AGW achieve the highest R-1 on RGBNT100 (95.6%) and MSVR310 (46.9%), respectively. On the large-scale RGBNT100 dataset, our FUSE method establishes a new SOTA mAP of 88.5%, representing an absolute improvement of 1.3% over the

second-best method (IDEA at 87.2%). Although our R-1 of 96.9% is slightly lower than DeMo, it remains competitive. On the challenging MSVR310 dataset, FUSE achieves full SOTA performance, leading both primary metrics. Our mAP reaches 50.1%, advancing the previous SOTA by 0.9% over DeMo (49.2%). Crucially, our R-1 accuracy of 65.7% outperforms the previous SOTA (IDEA at 62.4%) by 3.3%. These gains validate FUSE's performance and generalization, attributing efficacy to our Frequency-domain Unification and Spectral Energy Alignment framework.

### 4.3. Ablation Study

To evaluate the contribution and interplay of each component within FUSE, we conduct ablation studies on the RGBNT201 dataset. The detailed performance metrics are summarized in Table 3.

**Ablation Study of Core Components.** To thoroughly evaluate the contribution and necessity of the CAM and the SDM, which are the core components of FUSE, we conduct comprehensive ablation studies on the RGBNT201 dataset. The detailed results are summarized in Table 3. Our baseline uses a shared Vision Transformer backbone, achieving 76.0% mAP and 77.8% R-1 accuracy.

**Cross-Modal Alignment Module.** Building upon the base framework, we first integrate the CAM. This integration enhances performance, boosting mAP by an absolute 1.2 % (to 77.2%) and significantly improving Rank-1 accuracy by 2.6 % (to 80.4%). This validates that CAM effectively enhances the overall model performance by enforcing fine-grained cross-modal feature alignment.

**Spectral Decomposition Module.** Next, we introduce the

*Table 7.* **Comparison with different $\mathcal{L}_{id}$, $\mathcal{L}_{tri}$ and $\mathcal{L}_{freq}$ for FUSE.** We show the best score in bold.

| $\mathcal{L}_{id}$ | $\mathcal{L}_{tri}$ | $\mathcal{L}_{freq}$ | mAP | R-1 |
|---|---|---|---|---|
| ✓ | ✗ | ✗ | 45.5 | 44.4 |
| ✗ | ✓ | ✗ | 74.2 | 77.9 |
| ✓ | ✓ | ✗ | 80.8 | 84.1 |
| ✓ | ✓ | ✓ | **81.4** | **86.1** |

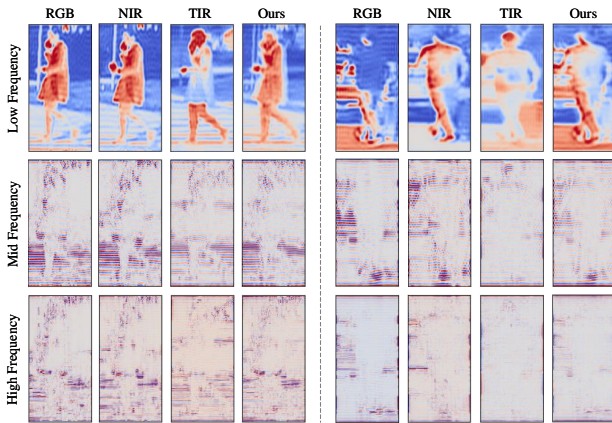

*Figure 4.* **Visualization of response distributions across frequency bands.** Band-wise responses of RGB, NIR, TIR, and ours. Low-frequency components are largely consistent across modalities, whereas mid and high-frequency bands exhibit stronger discrepancies and artifacts. Our method produces coherent mid and high-frequency structures with less modality-specific noise, yielding a stable multi-frequency representation.

SDM to the base framework. The SDM demonstrates an even more substantial contribution, achieving a strong mAP of 79.7% and R-1 of 82.1%. This represents a massive gain of 3.7% in mAP and 4.3 % in R-1 over the baseline. This result strongly validates that SDM is the dominant component, enhancing accuracy by effectively disentangling and processing features in the frequency domain, which is crucial for robust identity-discriminative feature extraction.

**FUSE.** Our complete FUSE model, integrating both CAM and SDM, achieves the optimal performance across all metrics, with 80.8% mAP and 84.1% R-1. This final configuration outperforms the best single-component model SDM only by an additional 1.1 % in mAP and 2.0 % in R-1. This final gain confirms the powerful synergistic effect between the CAM (alignment) and SDM (decomposition), demonstrating that the combination of frequency-domain processing and cross-modal alignment mechanisms is necessary to achieve the ultimate SOTA performance.

### 4.4. More Analysis

**Performance analysis under different frequency for SDM.** We analyze the weight distribution of the Low, Mid, and High-frequency components in Table 4. Using only a single band, the Low-frequency setting (1, 0, 0) performs best with 75.4% mAP and 78.7% R-1, reflecting its role in capturing global identity cues. Mid and High-frequency bands perform notably worse, showing that any single band is insufficient for robust cross-modal discrimination. Combining all three bands yields a clear performance gain, with the equal-weight configuration (0.3, 0.3, 0.3) achieving 79.7% mAP and 82.1% R-1, surpassing the best single-band result by 4.3% mAP. These results confirm that SDM benefits from jointly leveraging complementary information across all spectral bands.

**Analysis of the $\mathcal{L}_{freq}$ Weight.** To evaluate the effect of the proposed Auxiliary Frequency Consistency Loss $\mathcal{L}_{freq}$ and determine its optimal weight $\lambda_{freq}$, we conduct a hyperparameter analysis summarized in Table 5. Without this loss ($\lambda_{freq} = 0.0$), the model achieves 80.8% mAP and 84.1% Rank-1. Introducing a moderate constraint ($\lambda_{freq} = 0.1$) yields the best performance, improving to 81.4% mAP and 86.1% Rank-1. Larger weights ($\lambda_{freq} = 0.2, 0.3$) lead to performance drops, indicating over-regularization that

suppresses modality-specific cues. These results show that balancing discriminative learning with spectral alignment is essential, and $\lambda_{freq} = 0.1$ is adopted as the optimal setting.

**Performance analysis different kernel size for FUSE.** We investigate the impact of different convolution configurations on model performance in Table 6. Adopting larger kernel sizes, such as the (7, 7, 7) configuration, results in suboptimal performance with 79.3% mAP and 82.8% R-1, suggesting that excessive receptive fields tend to over-smooth fine-grained spectral details. Similarly, hybrid configurations with varying scales (e.g., 3, 5, 7) fail to yield improvement, indicating that complex spatial biases do not necessarily enhance feature distinctiveness. In contrast, the uniform small kernel configuration (3, 3, 3) achieves the optimal performance of 81.4% mAP and 86.1% R-1, surpassing the large-kernel setting by 2.1% mAP. These results confirm that a compact and consistent spatial inductive bias is most effective for preserving the structural integrity of disentangled frequency representations.

**Analysis of Loss Functions.** To evaluate each supervision term, we conduct ablation experiments on $\mathcal{L}_{id}$, $\mathcal{L}_{tri}$, and $\mathcal{L}_{freq}$. Table 7 shows that standard $\mathcal{L}_{id}$ and $\mathcal{L}_{tri}$ establish a solid baseline (80.8% mAP, 84.1% Rank-1). By incorporating $\mathcal{L}_{freq}$, FUSE achieves 81.4% mAP and 86.1% Rank-1, yielding gains of 0.6% and 2.0%. This enhancement validates that $\mathcal{L}_{freq}$ facilitates cross-modal spectral alignment by providing complementary characteristics overlooked by spatial losses. These gains confirm that spectral consistency is essential for robust multi-modal learning, ensuring identity-discriminative detail and stability.

**Visualization of responses in different frequency bands.** To qualitatively examine band-wise behavior across modal-

ities, we visualize the low-frequency, mid-frequency, and high-frequency responses of RGB, NIR, TIR, and our output, across two representative test cases. The low-frequency maps are relatively consistent across modalities, mainly reflecting the pedestrian silhouette and global intensity layout. In contrast, the mid and high-frequency maps reveal stronger modality-specific patterns, including noticeable cross-modal discrepancies and stripe-like artifacts. Compared with individual modalities, our method produces cleaner and more coherent structures in the mid and high bands while suppressing modality-dependent noise, resulting in a more stable multi-frequency representation.

## 5. Conclusion

In this paper, we propose **FUSE**, a novel framework for multi-modal object ReID that enhances spectral representation and robustness through frequency-domain disentanglement and alignment. Specifically, we introduce the Spectral Decomposition Module (SDM) to adaptively partition features into low, mid, and high-frequency subspaces, thereby ensuring complete spectral representation. Following this, the Cross-Modal Alignment Module (CAM) enforces fine-grained cross-modal feature alignment to effectively mitigate heterogeneity. Extensive experiments on the RGBNT201, RGBNT100, and MSVR310 datasets demonstrate the effectiveness and efficiency of our model.

## Acknowledgments

This work was supported in part by the National Natural Science Foundation of China under Grant U23A20276, and in part by the Guangdong Basic and Applied Basic Research Foundation under Grant 2024A1515110260.

## Impact Statement

This paper presents work whose goal is to advance the field of Machine Learning. There are many potential societal consequences of our work, none of which we feel must be specifically highlighted here.

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

# A. Supplemental Material

## A.1. Evaluation on Missing-Modality Scenarios

In real-world applications, sensor limitations or environmental factors often lead to incomplete modality inputs, making model robustness under these conditions a critical concern for deployment. Following the standard evaluation protocol, we assess the performance of FUSE under all six possible missing-modality configurations on the RGBNT201 dataset, with detailed results presented in Table 8. The results confirm that single-spectral methods suffer from severe performance degradations when image spectra are missing; for instance, the best single-spectral method, PCB, only achieves an average mAP of 19.7%. Multi-modal methods demonstrate better intrinsic robustness, with TOP-ReID setting a strong baseline, achieving an average mAP of 44.4% and R-1 45.4%. However, our proposed FUSE consistently and substantially outperforms all competitors in these challenging scenarios, achieving the new SOTA in all six configurations and the average metrics. FUSE reaches an average mAP of 49.9%, demonstrating a significant absolute gain of 5.5% over TOP-ReID. Our average R-1 also reaches 50.7%, surpassing TOP-ReID by 5.3%. This superior resilience validates FUSE's robustness and intrinsic capability, achieved without specialized modules, directly owing to our Frequency-domain Unification and Spectral Energy Alignment framework.

*Table 8.* **Performance of missing-modality settings on RGBNT201.** "M (X)" means missing the X image modality. The best and second results are in bold and underlined, respectively.

| | Methods | M (RGB) | | M (NIR) | | M (TIR) | | M (RGB+NIR) | | M (RGB+TIR) | | M (NIR+TIR) | | Average | |
|---|---|---|---|---|---|---|---|---|---|---|---|---|---|---|---|
| | | mAP | R-1 | mAP | R-1 | mAP | R-1 | mAP | R-1 | mAP | R-1 | mAP | R-1 | mAP | R-1 |
| Single | HACNN (Li et al., 2018) | 12.5 | 11.1 | 20.5 | 19.4 | 16.7 | 13.3 | 9.2 | 6.2 | 6.3 | 2.2 | 14.8 | 12.0 | 13.3 | 10.7 |
| | MUDeep (Qian et al., 2017) | 19.2 | 16.4 | 20.0 | 17.2 | 18.4 | 14.2 | 13.7 | 11.8 | 11.5 | 6.5 | 12.7 | 8.5 | 15.9 | 12.9 |
| | OSNet (Zhou et al., 2019) | 19.8 | 17.3 | 21.0 | 19.0 | 18.7 | 14.6 | 12.3 | 10.9 | 9.4 | 5.4 | 13.0 | 10.2 | 15.7 | 12.9 |
| | MLFN (Chang et al., 2018) | 20.2 | 18.9 | 21.1 | 19.7 | 17.6 | 11.1 | 13.2 | 12.1 | 8.3 | 3.5 | 13.1 | 9.1 | 15.6 | 12.4 |
| | CAL (Rao et al., 2021) | 21.4 | 22.1 | 24.2 | 23.6 | 18.0 | 12.4 | 18.6 | 20.1 | 10.0 | 5.9 | 17.2 | 13.2 | 18.2 | 16.2 |
| | PCB (Sun et al., 2018) | 23.6 | 24.2 | 24.4 | 25.1 | 19.9 | 14.7 | 20.6 | 23.6 | 11.0 | 6.8 | 18.6 | 14.4 | 19.7 | 18.1 |
| Multi | PFNet (Zheng et al., 2021) | - | - | 31.9 | 29.8 | 25.5 | 25.8 | - | - | - | - | 26.4 | 23.4 | - | - |
| | DENet (Zheng et al., 2023a) | - | - | 35.4 | 36.8 | 33.0 | 35.4 | - | - | - | - | 32.4 | 29.2 | - | - |
| | TOP-ReID (Wang et al., 2024a) | 54.4 | 57.5 | 64.3 | 67.6 | 51.9 | 54.5 | 35.3 | 35.4 | 26.2 | 26.0 | 34.1 | 31.7 | 44.4 | 45.4 |
| | DESANet (Dong et al., 2025) | 54.1 | 55.1 | 65.0 | 67.3 | 56.6 | **56.6** | 35.7 | 38.2 | **28.5** | **28.8** | 34.2 | 29.8 | 45.7 | 46.0 |
| | **FUSE** | **62.7** | **64.0** | **70.9** | **73.7** | **58.0** | 55.1 | **39.6** | **42.3** | 26.4 | 26.6 | **41.5** | **42.7** | **49.9** | **49.9** |

## A.2. Computational Cost and Efficiency

Table 9 summarizes the computational cost of FUSE and representative recent methods in terms of parameters and FLOPs. FUSE attains the best efficiency profile, achieving the lowest cost on both metrics with 59.3M parameters and 24.7G FLOPs. Compared with HTT, FUSE reduces the parameter count from 85.6M to 59.3M and decreases FLOPs from 33.1G to 24.7G, indicating a substantially lighter architecture. The gap is more pronounced relative to EDITOR, where FUSE nearly halves the parameters (117.5M vs. 59.3M) and cuts computation by a large margin (38.6G vs. 24.7G). While TOP-ReID achieves competitive FLOPs, it remains markedly less parameter-efficient, requiring 278.2M parameters with 34.5G FLOPs. Overall, these results demonstrate that FUSE delivers frequency-domain modeling and cross-modal alignment with significantly reduced model size and computation, making it more practical for scalable training and deployment.

*Table 9.* **Comparison of computational cost with methods.** We show the best result in bold.

| Methods | Params(M) | Flops(G) |
|---|---|---|
| HTT (Wang et al., 2024b) | 85.6 | 33.1 |
| EDITOR (Zhang et al., 2024) | 117.5 | 38.6 |
| TOP-ReID (Wang et al., 2024a) | 278.2 | 34.5 |
| DeMo (Wang et al., 2025b) | 98.8 | 35.1 |
| MambaPro (Wang et al., 2025a) | 160.3 | 25.2 |
| DESANet (Dong et al., 2025) | 256.9 | 33.1 |
| Ours | **59.3** | **24.7** |

