# OpenReview forum: "FUSE: Frequency-domain Unification and Spectral Energy Alignment for Multi-modal Object Re-Identification"
_ICML.cc/2026/Conference — ICML 2026 regular_

### Official Review · Reviewer_npZu · 2026-02-26

**Soundness:** 4
**Presentation:** 4
**Significance:** 4
**Originality:** 4
**Overall Recommendation:** 5
**Confidence:** 5

**Summary:**

This work introduces FUSE, which addresses the ‘low-frequency bias’ and unstable cross-modal alignment in multi-modal ReID via a two-stage frequency-domain framework. The Spectral Decomposition Module (SDM) adaptively disentangles features into low, mid, and high-frequency subspaces using learnable filters and band-specific convolutions to capture hierarchical spectral semantics. The Cross-Modal Alignment Module (CAM) subsequently enforces spectral energy alignment and subspace complementarity across RGB, NIR, and TIR modalities through frequency-aware attention and consistency regularization.

**Compliance With Llm Reviewing Policy:**

Affirmed.

**Final Justification:**

The authors have satisfactorily addressed my concerns in the rebuttal. In particular, they clarified the efficiency–performance trade-off more explicitly, corrected the presentation issues, and strengthened the discussion of related frequency-domain methods and the advantages of their approach. Overall, I continue to view this as a technically solid and clearly motivated work with strong empirical results, and I therefore support acceptance.

**Key Questions For Authors:**

1. There are still several typos: The caption in Figure3, ‘Feed Forword’ -> ‘Feed Forward’.
2. While Table 8 in the supplemental material provides some comparisons of Parameters and FLOPs, the main text lacks a comprehensive discussion relating these efficiency metrics to the recognition performance (mAP, Rank-1) across different methods. Can the authors provide a more detailed comparison of computations with different methods?
3. The related work section lacks sufficient comparisons with recent frequency-domain methods, so the authors should supplement such comparisons and clearly clarify the advantages of their approach.

**Limitations:**

yes

**Strengths And Weaknesses:**

**Strengths**:

1.The proposed framework is well-structured, with clearly defined motivations centered on overcoming the inherent low-frequency bias in multi-modal networks. The integration of the Spectral Decomposition Module (SDM) and the Cross-Modal Alignment Module (CAM) effectively enhance spectral feature interaction and cross-modal energy alignment, addressing the instability of existing spatial-domain methods.

2.The performance of this work is quite strong, achieving significant improvements over the previous SOTA, which provides solid empirical evidence for the effectiveness of the frequency-domain paradigm.

**Weaknesses**:

1.Inconsistent and Delayed Introduction of Technical Terms: There exist redundant or delayed definitions for technical terms (e.g., ViT, DFT), which are either repeated or explained only after frequent use.

2.Frequency Mask Explanation is Overly Verbose: The explanation of learnable radial masks spans nearly 0.5 pages. The derivation of radial distance and partitioning logic can be simplified or moved to the supplementary material. The current version distracts from the main technical innovation: the adaptive frequency-specific convolution strategy.

---

> ### Author Rebuttal · Authors · 2026-03-28
>
> > Q1: There are still several typos: The caption in Figure3, ‘Feed Forword’ -> ‘Feed Forward’.
> >
>
> A1: Thank you for pointing this out. We have corrected the typo in Figure 3 and performed a comprehensive review of the manuscript to ensure no similar issues remain.
>
> > Q2: While Table 8 in the supplemental material provides some comparisons of Parameters and FLOPs, the main text lacks a comprehensive discussion relating these efficiency metrics to the recognition performance (mAP, Rank-1) across different methods. Can the authors provide a more detailed comparison of computations with different methods?
> >
>
> A2: Thank you for the insightful comment. We agree that a clearer connection between computational cost and recognition performance is important. We have revised and extended Table 8 to provide a more explicit efficiency–performance comparison.
>
> Existing methods show that higher computational cost does not necessarily translate to better performance. For example, **HTT (85.6M / 33.1G)** achieves 71.1% mAP / 73.4% R-1, while **EDITOR (117.5M / 38.6G)** further increases complexity but drops to 66.5% mAP / 68.3% R-1. **TOP-ReID (278.2M / 34.5G)** improves performance to 72.3% mAP / 76.6% R-1, but at the cost of extremely large parameter overhead, indicating relatively low parameter efficiency.
>
> In contrast, **FUSE (59.3M / 24.7G)** achieves **81.4% mAP / 86.1% R-1**, delivering the best performance with the lowest computational cost. Specifically, FUSE reduces FLOPs by ~25% compared to **HTT** while achieving **+10.3% mAP / +12.7% R-1** gains, uses ~50% fewer parameters than **EDITOR** with a **+14.9% mAP** improvement, and reduces parameters by ~4.7× compared to **TOP-ReID** while improving **+9.1% mAP / +9.5% R-1**. These results demonstrate a clearly superior efficiency–performance trade-off.
>
> This advantage stems from architectural design. **HTT** relies on multi-stage Transformer interactions, **EDITOR** adopts multi-branch frequency modeling, and **TOP-ReID** uses repeated cross-attention, all leading to redundant computation. In contrast, **FUSE** employs a shared backbone, lightweight spectral decomposition (SDM), and single-round cross-modal alignment (CAM), achieving more efficient representation learning.
>
> We will revise Section 4.4 of the main text to incorporate this analysis.
>
> | Methods | Params(M) | Flops(G) | mAP | R-1 | R-5 | R-10 | Average |
> | --- | --- | --- | --- | --- | --- | --- | --- |
> | HTT | 85.6 | 33.1 | 71.1 | 73.4 | 83.1 | 87.3 | 78.73 |
> | EDITOR | 117.5 | 38.6 | 66.5 | 68.3 | 81.1 | 88.2 | 76.03 |
> | TOP-ReID | 278.2 | 34.5 | 72.3 | 76.6 | 84.7 | 89.4 | 80.75 |
> | FUSE(Ours) | 59.3 | 24.7 | 81.4 | 86.1 | 91.5 | 93.8 | 88.20 |
>
> > Q3: The related work section lacks sufficient comparisons with recent frequency-domain methods, so the authors should supplement such comparisons and clearly clarify the advantages of their approach.
> >
>
> A3: Thank you for the valuable feedback. We agree that the related work section would benefit from a more comprehensive comparison with recent frequency-domain methods. In the revised manuscript, we have extended this section to include representative approaches such as **FDNM, MFENet, and EDITOR**, and provide a clearer technical comparison. Specifically, we now discuss their different frequency modeling strategies, including **fixed Fourier decomposition (FDNM)**, **coarse low/high-frequency splitting (MFENet)**, and **multi-scale wavelet-based modeling (EDITOR)**. We further clarify their limitations in terms of **incomplete spectral decomposition**, **lack of explicit cross-modal alignment**, or **high computational overhead.** If you have any recommendations, we welcome further suggestions or references.

---

> > ### Author Rebuttal · Reviewer_npZu · 2026-04-02
> >
> > The authors have satisfactorily addressed all my concerns. I maintain my recommendation: 5 (Accept).

---

> > > ### Author Response · Authors · 2026-04-03
> > >
> > > We sincerely appreciate your positive feedback. We are pleased that our response has addressed your concerns, and we are grateful for your recognition and support of our work.

---

### Official Review · Reviewer_Zw63 · 2026-03-07

**Soundness:** 3
**Presentation:** 3
**Significance:** 3
**Originality:** 3
**Overall Recommendation:** 5
**Confidence:** 3

**Summary:**

Existing multimodal re-identification methods overemphasize low-frequency attributes such as color, illumination, and coarse appearance, neglecting mid- and high-frequency structures that encode geometric, textural, and identity-distinguishing details. This imbalance leads to incomplete spectral representation and unstable cross-modal alignment. To address this, this paper proposes FUSE, a frequency domain framework that enhances robustness under varying illumination and sensor conditions through a two-stage process: spectral decoupling and energy alignment.

**Compliance With Llm Reviewing Policy:**

Affirmed.

**Final Justification:**

The author has resolved my concerns. Overall, this paper represents technically sound, experimentally rigorous, and clearly motivated work; I support its acceptance.

**Key Questions For Authors:**

Q1：According to the visualization results in Figure 4, the superiority of the proposed method does not appear very evident. Could the authors provide further explanation or analysis regarding these results?

Q2：The motivation of the paper is based on the observation that existing methods tend to overlook low-frequency attributes. However, the potential negative impact of this issue is not strongly demonstrated, especially in the experimental section. Could the authors provide further evidence or discussion?

Q3：The paper emphasizes that spatial-domain models exhibit a low-frequency bias. Could the authors provide an analysis of the contribution of different frequency bands to the final recognition accuracy, along with corresponding experimental validation?

Q4：In the objective function, besides the frequency consistency loss, what are the mathematical formulations of the other two losses mentioned in the training objective? It would be helpful if the authors could provide their explicit definitions.

Q5：The ablation study does not appear to analyze the individual contributions of the three loss functions. Could the authors provide experiments to verify the effect of each loss component?

**Limitations:**

From a societal perspective, since multi-modal ReID systems are closely related to surveillance and identity tracking applications, it would be beneficial to briefly acknowledge possible concerns regarding privacy, misuse in large-scale monitoring, or deployment without proper governance. A short discussion on responsible usage, data governance, or ethical considerations could strengthen the impact statement.

**Strengths And Weaknesses:**

Strengths：

1)The paper presents a clear and well-motivated research problem, highlighting the limitations of existing multi-modal ReID methods that tend to focus primarily on low-frequency information.

2)The proposed framework introduces a structured modular design, including the Spectral Decomposition Module (SDM) and Cross-Modal Alignment Module (CAM), which are logically organized.

3)The methodological pipeline is described clearly, and the overall presentation of the framework is relatively easy to follow, resulting in good readability.

Weaknesses

1)The formatting of references is inconsistent throughout the manuscript. For example, some conference or journal names are written in uppercase while others appear in lowercase.

2)The theoretical depth of the work appears somewhat limited. The proposed approach mainly relies on combining several modules, and the novelty is largely reflected in the integration of these components rather than in the development of a fundamentally new frequency-domain theoretical mechanism.

3)In the objective formulation, only the frequency consistency loss is explicitly described, while the mathematical forms of the other losses mentioned in the training objective are not clearly provided.

4)Although efficiency is briefly mentioned in the appendix, the paper does not provide a detailed or systematic analysis of computational complexity or runtime efficiency.

---

> ### Author Rebuttal · Authors · 2026-03-28
>
> > Q1: Figure 4 does not clearly demonstrate the superiority of the proposed method; could the authors provide further analysis or explanation?
> >
>
> A1: Thank you for your comment. The superiority of FUSE in Figure 4 is mainly reflected in the mid- and high-frequency bands, which capture fine-grained structural details for cross-modal alignment. In the mid-frequency bands, original modalities show strong artifacts such as horizontal stripe patterns (e.g., RGB and NIR). FUSE suppresses these artifacts and produces more coherent structural responses. In the high-frequency bands, some modalities (e.g., TIR) lose fine details and appear nearly blank, while FUSE recovers complementary structural information, yielding clearer outlines without introducing excessive noise.
>
> > Q2: The motivation of the paper is based on the observation that existing methods tend to overlook low-frequency attributes. However, the potential negative impact of this issue is not strongly demonstrated, especially in the experimental section. Could the authors provide further evidence or discussion?
> >
>
> A2: Thank you for the feedback. Our motivation is based on the observation that spatial-domain methods overly rely on low-frequency cues while under-utilizing mid- and high-frequency information with finer structural and texture details. As shown in Table 4, the low-frequency component alone achieves 75.4% mAP, while incorporating mid- and high-frequency information improves performance to 79.7% (+4.3%), revealing a clear bottleneck when these components are ignored. This is further supported by Figure 4, where mid and high-frequency bands exhibit strong cross-modal artifacts (e.g., stripe noise), leading to unstable feature alignment. We will further emphasize this analysis in the revised manuscript.
>
> > Q3: The paper argues that existing methods overlook low-frequency attributes, but the negative impact of this issue is not sufficiently demonstrated, especially in the experiments. Could the authors provide stronger evidence or analysis?
> >
>
> A3: Thank you for the feedback. We analyze the contribution of different frequency bands in Table. The low-frequency component achieves the best single-band performance (75.4% mAP), highlighting its role in capturing global structure and indicating that the baseline mainly relies on low-frequency information. In comparison, mid- and high-frequency components perform lower (72.2% and 69.0% mAP), as they capture partial structure and texture details. However, combining all bands improves performance to 79.7% mAP, showing that mid- and high-frequency features provide complementary discriminative information. These results confirm that although spatial models exhibit a low-frequency bias, incorporating mid- and high-frequency components is crucial for optimal recognition.
>
> | Methods | mAP | R-1 |
> | --- | --- | --- |
> | low-frequency | 75.4 | 78.7 |
> | mid-frequency | 72.2 | 73.4 |
> | high-frequency | 69.0 | 73.8 |
> | Ours (low+mid+high) | 79.7 | 82.1 |
>
> > Q4: In the objective function, what are the explicit mathematical formulations of the other two losses besides the frequency consistency loss?
> >
>
> A4: Thank you for your suggestion. In addition to the frequency consistency loss, we employ standard identity classification and metric learning objectives in ReID.
> The **identity loss** treats Re-ID as a classification problem and is implemented using cross-entropy with label smoothing:
>
> $\mathcal{L}{id} = - \frac{1}{n} \sum_{i=1}^{n} \log(p(y_i \mid x_i)),$
>
> where $n$ is the batch size, and $p(y_i|x_i)$ is the predicted probability of input $x_i$ belonging to its ground-truth class $y_i$.
> The **triplet loss** is formulated as:
>
> $\mathcal{L}{tri}(i, j, k) = \max(\rho + d{ij} - d_{ik}, 0),$
>
> where $d(\cdot)$ measures the Euclidean distance between the anchor ($i$), positive ($j$), and negative ($k$) samples.
>
> > Q5: The ablation study does not analyze the individual contributions of the three loss functions; could the authors provide experiments to verify each component’s effect?
> >
>
> Thank you for your feedback. As shown, the standard combination of $\mathcal{L}{id}$ and $\mathcal{L}{tri}$ establishes a strong baseline (80.8% mAP). Integrating our proposed $\mathcal{L}{freq}$ further boosts performance to 81.4% mAP and 86.1% Rank-1. This confirms that $\mathcal{L}{freq}$ effectively provides complementary gains by enforcing cross-modal spectral alignment. We will include this detailed loss ablation table in the revised manuscript.
>
> | $\mathcal{L}_{id}$ | $\mathcal{L}_{tri}$ | $\mathcal{L}_{freq}$ | **mAP** | **R-1**  |
> | --- | --- | --- | --- | --- |
> | $\checkmark$ |  |  | 45.5 | 44.4 |
> |  | $\checkmark$ |  | 74.2 | 77.9 |
> | $\checkmark$ | $\checkmark$ |  | 80.8 | 84.1 |
> | $\checkmark$ | $\checkmark$ | $\checkmark$ | **81.4** | **86.1** |

---

> > ### Author Rebuttal · Reviewer_Zw63 · 2026-04-01
> >
> > All my concerns have been resolved, so I will raise my rating to 5

---

> > > ### Author Response · Authors · 2026-04-01
> > >
> > > We sincerely appreciate your positive feedback. We are pleased that our response has addressed your concerns, and we are grateful for your recognition and support of our work.

---

### Official Review · Reviewer_cx6T · 2026-03-07

**Soundness:** 2
**Presentation:** 3
**Significance:** 2
**Originality:** 2
**Overall Recommendation:** 4
**Confidence:** 5

**Summary:**

The paper proposes a frequency-domain framework for multi-modal object re-identification that utilizes a Spectral Decomposition Module to partition features into low, mid, and high-frequency subspaces and a Cross-Modal Alignment Module for spectral energy alignment. The method achieves state-of-the-art performance on public datasets.

**Compliance With Llm Reviewing Policy:**

Affirmed.

**Final Justification:**

Thanks for the response. The authors have largely addressed my concerns, but I still find the novelty to be limited. Therefore, I am raising my score to 4.

**Key Questions For Authors:**

Please refer to the weaknesses.

**Limitations:**

Please refer to the weaknesses.

**Strengths And Weaknesses:**

Strengths:
1. The proposed model achieves significant performance on the RGBNT201 dataset.
2. The introduced Spectral Decomposition Module allows for adaptive frequency partitioning using learnable radial masks, providing a more flexible approach than fixed-threshold band definitions.
3. The architecture appears more parameter-efficient than some previous SOTA models.

Weaknesses:
1. The novelty is limited because the integration of frequency analysis is a well-explored area in multimodal fusion tasks, and the proposed alignment strategy relies on the cross-attention module, which is a typical method.
2. The computational efficiency analysis in Table 8 is misleadingly narrow, as it only compares methods from 2024 while omitting the latest models.
3. The paper fails to provide inference time, which is a critical omission for a method involving repeated 2D DFT and IDFT operations that incur significant hardware scheduling overhead.
4. The effectiveness of the proposed frequency consistency loss is highly sensitive to the lambda, with performance dropping significantly when the weight is increased beyond the optimal value of 0.1. Please provide an explanation and analysis.
5. The proposed model achieves SOTA mAP on the RGBNT100, but Rank-1 is lower than DeMo. Please discuss this phenomenon.

---

> ### Author Rebuttal · Authors · 2026-03-28
>
> > Q1: The novelty is limited since frequency analysis is well explored in multimodal fusion and the alignment strategy relies on standard cross-attention.
> >
>
> A1: We thank the reviewer for the constructive comment. While frequency analysis and cross-attention are established, **the novelty of FUSE lies in their systematic integration for multi-spectral (RGB–NIR–TIR) ReID**:
>
> **1. Adaptive and Heterogeneous Spectral Decomposition (SDM):** Our SDM introduces **adaptive frequency partitioning** and assigns Dilated, Standard, and Depthwise convolutions to low-, mid-, and high-frequency subspaces, enabling a **heterogeneous design for spectral disentanglement.**
>
> **2. Spectral Energy Alignment via $\mathcal{L}_{freq}$ (CAM):** Instead of standard spatial alignment, CAM performs **symmetric all-to-one aggregation**, supervised by **Frequency Consistency Loss ($\mathcal{L}_{freq}$)** to align spectral energy across heterogeneous sensors, differing from typical cross-attention.
>
> We will revise the manuscript to clarify these distinctions and emphasize our novelties.
>
> > Q2: The computational efficiency analysis in Table 8 is misleadingly narrow, as it only compares methods from 2024 while omitting the latest models.
> >
>
> A2: We thank the reviewer for the comment. We will extend Table 8 to include more recent methods for a more comprehensive comparison. To address this concern, we have additionally compared FUSE with several recent models (e.g., DeMo, MambaPro, and DESANet). As shown in the updated results, FUSE achieves 59.3M parameters and 24.7G FLOPs, which remains highly competitive compared to these latest approaches (e.g., DeMo: 98.8M / 35.1G, DESANet: 256.9M / 33.1G). These results further demonstrate that FUSE maintains a favorable efficiency–performance trade-off. We will include these comparisons in the revised manuscript.
>
> | Methods | Venue | Params(M) | Flops(G) |
> | --- | --- | --- | --- |
> | DeMo | AAAI 2025 | 98.8 | 35.1 |
> | MambaPro | AAAI 2025 | 160.3 | 25.2 |
> | DESANet | TIP 2025 | 256.9 | 33.1 |
> | FUSE (Ours) | - | 59.3 | 24.7 |
>
> > Q3: The paper fails to provide inference time, which is a critical omission for a method involving repeated 2D DFT and IDFT operations that incur significant hardware scheduling overhead.
> >
>
> A3: We thank the reviewer for this important comment. We will include inference time in the revised manuscript for a more complete evaluation. Empirically, FUSE processes the test set in **11.3 seconds**, faster than TOP-ReID (13.7s), EDITOR (16.1s), and DESANet (12.4s), while achieving superior performance (**81.4% mAP / 86.1% Rank-1**) with lower cost (**24.7G MACs, 59.3M parameters**). This shows the frequency-domain design introduces no practical runtime overhead and achieves a strong efficiency–performance trade-off, mainly due to FFT-based operations on compact features with optimized  $\mathcal{O}(N \log N)$ implementations. We will include these results in the revised manuscript
>
> | Methods | Test Time(s) | Mac(G) | Params(M) | mAP(%) | R-1(%) |
> | --- | --- | --- | --- | --- | --- |
> | baseline (TOP-REID) | 13.7 | 34.5 |  278.2 | 72.3 | 76.6 |
> | EDITOR | 16.1 | 38.6 | 117.5 | 65.7 | 68.8 |
> | DESANet | 12.4 | 33.1 | 256.9 | 74.6 | 77.6 |
> | FUSE(Ours) | 11.3 | 24.7 |  59.3 |  81.4 | 86.1 |
>
> > Q4: The effectiveness of the proposed frequency consistency loss is highly sensitive to the lambda, with performance dropping significantly when the weight is increased beyond the optimal value of 0.1. Please provide an explanation and analysis.
> >
>
> A4: Thank you for your feedback. The sensitivity to $\lambda_{freq}$ reflects the balance between spectral alignment and discriminative feature learning. When $\lambda_{freq}$ is set to a moderate value (e.g., 0.1), the loss improves cross-modal spectral consistency. However, larger weights lead to over-regularization, which suppresses modality-specific cues and degrades performance. This behavior is consistent with the trade-off between alignment and discrimination in multi-modal learning. Based on our analysis, $\lambda_{freq}=0.1$ provides the best balance and is adopted in our final model.
>
> > Q5: The proposed model achieves SOTA mAP on the RGBNT100, but Rank-1 is lower than DeMo. Please discuss this phenomenon.
> >
>
> A5: Thank you for the insightful comment. This reflects a trade-off between Rank-1 accuracy and overall retrieval quality (mAP). DeMo emphasizes spatial deformable aggregation, benefiting precise local alignment and improving top-1 matching. In contrast, FUSE leverages frequency-domain spectral alignment to capture holistic structural information, enhancing cross-modal consistency and leading to better separation across the ranking list, thus improving mAP. This trade-off is common in ReID systems. Notably, the slight Rank-1 drop occurs only on RGBNT100. On other benchmarks (e.g., RGBNT201, MSVR310), FUSE achieves state-of-the-art performance in both mAP and Rank-1, demonstrating strong generalizability.

---

> > ### Author Rebuttal · Reviewer_cx6T · 2026-04-04
> >
> > Thanks for the response. The authors have largely addressed my concerns, but I still find the novelty to be limited. Therefore, I am raising my score to 4.

---

> > > ### Author Response · Authors · 2026-04-04
> > >
> > > We sincerely appreciate your positive feedback. We are pleased that our response has addressed your concerns, and we are grateful for your recognition and support of our work.

---

### Official Review · Reviewer_dkEY · 2026-03-10

**Soundness:** 3
**Presentation:** 2
**Significance:** 3
**Originality:** 2
**Overall Recommendation:** 4
**Confidence:** 3

**Summary:**

This paper proposes FUSE, a frequency-domain framework for multi-modal object re-identification across RGB, NIR, and TIR inputs. The method combines 1) a Spectral Decomposition Module, which splits features into low, mid, and high-frequency components 2) a Cross-Modal Alignment Module, which aligns information across modalities using cross-attention. Experiments on RGBNT201, RGBNT100, and MSVR310 report improved mAP and Rank-1 performance over prior methods.

**Compliance With Llm Reviewing Policy:**

Affirmed.

**Final Justification:**

All my concerns have been resolved and I would like to improve rating to 4.

**Key Questions For Authors:**

See weaknesses.

**Limitations:**

yes

**Strengths And Weaknesses:**

# strengths

- First reformulates multi-modal ReID as a two-stage process of spectral disentanglement and energy alignment. Via the SDM, it adaptively partitions low, mid, and high-frequency subspaces, overcoming the limitation of existing methods that overly rely on low-frequency cues.
- The empirical results are strong on the reported benchmarks. In Table 1, FUSE improves RGBNT201 performance to 81.4 mAP / 86.1 R-1, and in Table 2 it achieves the best mAP on RGBNT100 and both best mAP/R-1 on MSVR310.
- With significantly fewer parameters (59.3M) and lower FLOPs (24.7G) than mainstream methods, FUSE maintains superior performance under missing-modality scenarios (average mAP 49.9%), demonstrating strong potential for real-world deployment.

# weakness

- Section 2 (Related Work) is presented as a single continuous block of text, resulting in unclear logical hierarchy and increased reading difficulty. It is recommended to split this section into 2-3 paragraphs to enhance readability.
- In Figure 3, the feed-forward neural network layer in the network architecture is misspelled as "Feed Forword" instead of the correct "Feed Forward".
- Figure 3 shows that each operation uses one modality as the Query and the other two as Key/Value. This design requires executing three independent Cross-Attention operations, potentially leading to computational redundancy.
- Some multi-modal re-id works are encouraged to be reviewed, such as llava-reid and chat-reid.
- Equation (7) employs a completely non-differentiable indicator function for hard truncation. The authors fail to clarify how gradients backpropagate through this step during end-to-end training.
- The paper highlights using dilated, standard, and depthwise convolutions for specific frequency bands as a core design. However, the ablation study (Table 6) only tests kernel sizes, failing to ablate the convolution types to prove this heterogeneous design is actually necessary.

#

---

> ### Author Rebuttal · Authors · 2026-03-28
>
> > Q1: Section 2 is presented as a single continuous block with unclear structure, and should be split into 2–3 paragraphs to improve readability.
> >
>
> A1: Thank you for pointing this out. We will revise Section 2 to improve clarity by restructuring it into thematic paragraphs. Specifically, it will be divided into subsections on CNN-based fusion, Transformer-based interaction, and frequency-domain modeling to improve logical flow. We welcome further suggestions or references.
>
> > Q2: In Figure 3, “Feed Forword” is misspelled and should be corrected to “Feed Forward.”
> >
>
> A2: Thank you for pointing this out. We have corrected the typo in Figure 3 and performed a comprehensive review of the manuscript to ensure no similar issues remain.
>
> > Q3: Figure 3 uses three separate cross-attention operations with different modalities as Query, which may introduce computational redundancy.
> >
>
> **A3:** Thank you for this insightful comment. Adding CAM improves performance (76.0 → 77.2 mAP, 77.8 → 80.4 Rank-1) with only +1.6M parameters, showing effective cross-modal interaction with minimal cost. Although CAM uses three cross-attention operations, they share parameters and run in parallel, avoiding redundancy while enabling complementary interactions. Compared to TPM in TOP-ReID (9.3M params, 67.8 mAP, 69.4 Rank-1), CAM achieves better performance with lower overhead. FUSE remains efficient with 59.3M parameters and 24.7G FLOPs, lower than TOP-ReID and HTT.
>
> | Methods | mAP | R-1 | Params(M) |
> | --- | --- | --- | --- |
> | baseline | 76.0 | 77.8 | 57.1 |
> | baseline +CAM | 77.2 | 80.4 | 58.7(+1.6) |
>
> > Q4: Some multi-modal re-id works are encouraged to be reviewed, such as llava-reid and chat-reid.
> >
>
> A4: Thank you for the suggestion. We will include LLaVA-ReID and ChatReID, and have added recent works [2–5]. Additional references can be incorporated if needed.
>
> [1] Lu Y, Lin Y, Yang M, et al. "Decoupled contrastive multi-view clustering with high-order random walks." *Proceedings of the AAAI conference on artificial intelligence*. 2024.
>
> [2] Lu, Yiding, et al. "LLaVA-ReID: Selective multi-image questioner for interactive person re-identification." *arXiv preprint arXiv:2504.10174* (2025).
>
> [3] Niu, Ke, et al. "Chatreid: Open-ended interactive person retrieval via hierarchical progressive tuning for vision language models." *Proceedings of the IEEE/CVF International Conference on Computer Vision*. 2025.
>
> [4] Niu, Ke, et al. "FDGReID: Federated Domain Generalization for Person Re-identification." *Machine Learning* 115.1 (2026): 22.
>
> [5] Yang M, Huang Z, Hu P, et al.  "Learning with twin noisy labels for visible-infrared person re-identification." *Proceedings of the IEEE/CVF conference on computer vision and pattern recognition*. 2022.
>
> > Q5: Equation (7) uses a non-differentiable indicator function for hard truncation but does not explain how gradients are backpropagated.
> >
>
> A5: We thank the reviewer for the comment on the non-differentiability of the indicator function in Equation (7). The hard masks  $\mathbb{I}[\cdot]$ are non-differentiable, raising concerns about gradient flow. However, the band boundaries are parameterized by a continuous vector $\boldsymbol{\theta}$, normalized via Softmax (Eq. 5), which defines $r_L, r_M, r_H$. During backpropagation, we adopt the Straight-Through Estimator (STE) to approximate gradients by routing them to $\boldsymbol{\theta}$ through the Softmax outputs. This enables differentiable learning of frequency boundaries without relaxing hard masks. We will revise Section 3.1 to clarify this strategy and include relevant references (e.g., Gumbel-Softmax, neural quantization).
>
> > Q6: The paper proposes using different convolution types for specific frequency bands, but the ablation only tests kernel sizes and does not verify the necessity of this heterogeneous design.
> >
>
> A6: We thank the reviewer for highlighting this gap. While Table 6 validates kernel sizes, it does not isolate convolution types across frequency bands. We conduct an additional ablation comparing our heterogeneous design with homogeneous baselines (all Conv, all DConv with d=2, and all DWConv). Our design outperforms all baselines; compared to the best (All DConv), it achieves +5.1% mAP and +7.4% Rank-1. This is because homogeneous designs impose mismatched inductive biases: Conv over-smooths high-frequency details, DConv suffers from gridding effects, and DWConv lacks channel interaction for low-frequency semantics. Aligning DConv, Conv, and DWConv with low-, mid-, and high-frequency bands enables complementary feature extraction.
>
> | Methods | mAP | R-1 | R-5 | R-10 | Average |
> | --- | --- | --- | --- | --- | --- |
> | DConv3x3, Conv3x3, DWConv3x3 | 81.4% | 86.1% | 91.5% | 93.8% | 88.2% |
> | Conv3x3, Conv3x3, Conv3x3 | 73.7% | 76.6% | 85.2% | 89.0% | 81.13% |
> | DConv3x3, DConv3x3, DConv3x3 | 76.3% | 78.7% | 87.7% | 91.3% | 83.50% |
> | DWConv3x3,  DWConv3x3, DWConv3x3 | 75.5% | 78.8% | 85.9% | 89.5% | 82.43% |

---

> > ### Author Rebuttal · Reviewer_dkEY · 2026-04-03
> >
> > Thanks for the response. I still have some questions about the novelty of this work. Could you further discuss the differences between SDM, CAM, and existing works.

---

> > > ### Author Response · Authors · 2026-04-03
> > >
> > > Thanks for your constructive feedback. We appreciate the reviewer’s follow-up. To clarify, the novelty of FUSE does not lie in the use of frequency transforms or attention mechanisms themselves, but in how these components are jointly formulated and coupled to address cross-modal discrepancies. We directly compare our design with existing methods below.
> > >
> > > **1. SDM vs. Existing Frequency-Domain Methods**
> > >
> > > Existing methods such as MFENet [1], FDNM [2], and EDITOR [3] use frequency information but rely on static and pre-defined decompositions, such as fixed low and high splits, amplitude and phase separation, or wavelets. In these methods, the frequency structure is not learned. In contrast, SDM introduces:
> > >
> > > (1) **Learnable radial partitioning:** SDM parameterizes frequency boundaries using a vector $θ$, allowing the model to adaptively learn the spectral partition through backpropagation instead of relying on fixed thresholds or heuristics.
> > >
> > > (2) **Frequency-aware inductive bias:** SDM assigns different operators, including dilated, standard, and depthwise convolutions, to low, mid, and high frequency bands respectively. This aligns the inductive bias with specific spectral characteristics such as global semantics and fine textures. Prior methods often apply homogeneous processing after decomposition and do not model this operator and frequency coupling.
> > >
> > > As a result, SDM is not a fixed frequency transform but a learnable and structured spectral representation mechanism.
> > >
> > > **2. CAM vs. Existing Cross-Modal Interaction**
> > >
> > > Existing methods such as TOP-ReID [4] and DeMo [5] mainly perform alignment in the spatial or token domain and often rely on sequential or relay-based interaction. CAM differs in two key aspects:
> > >
> > > (1) **Spectral-level alignment:** CAM explicitly aligns frequency energy distributions across modalities through the frequency consistency loss $\mathcal{L}_{freq}$, instead of only aligning feature embeddings. This directly addresses spectral discrepancies across sensors.
> > >
> > > (2) **Symmetric all-to-one interaction:** CAM constructs a joint Key and Value from all other modalities, for example $S_{NT}$ for the RGB stream, and performs alignment in a single step. This allows each modality to access the full multi-modal context directly and avoids the information bottlenecks caused by sequential propagation.
> > >
> > > **3. Conceptual Novelty**
> > >
> > > The key novelty lies in the coupled formulation of representation and alignment:
> > >
> > > (1) **From implicit to explicit modeling:** FUSE shifts from implicit spatial correlation to explicit spectral modeling and reasoning.
> > >
> > > (2) **Frequency as a learnable representation axis:** SDM transforms frequency from a fixed transform into a learnable and structured representation space.
> > >
> > > (3) **Alignment within the same space:** CAM performs cross-modal alignment within the spectral space defined by SDM, instead of aligning raw spatial features.
> > >
> > > In summary, SDM introduces adaptive spectral decomposition with frequency-aware inductive bias, while CAM performs spectral-level alignment with global interaction. Their combination establishes a unique frequency-domain formulation of multi-modal ReID, rather than a straightforward combination of existing frequency modeling and cross-modal attention mechanisms. This formulation is not only conceptually distinct but also empirically effective, achieving 81.4 mAP and 86.1 Rank-1 on RGBNT201, outperforming recent methods such as EDITOR (66.5 mAP), TOP-ReID (72.3 mAP), DESANet (74.6 mAP), and DeMo (79.0 mAP).
> > >
> > > | Methods | Venue | mAP | R-1 |
> > > | --- | --- | --- | --- |
> > > | EDITOR [3] | CVPR 2024 | 66.5 | 68.3 |
> > > | TOP-ReID [4] | AAAI 2024 | 72.3 | 76.6 |
> > > | DESANet [7] | TIP 2025 | 74.6 | 77.6 |
> > > | DeMo [5] | AAAI 2025 | 79.0 | 82.3 |
> > > | MambaPro [6] | AAAI 2025 | 78.9 | 83.4 |
> > > | FUSE (Ours) | - | 81.4 | 86.1 |
> > >
> > > [1] Discovering Multi-Frequency Embedding for Visible-Infrared Person Re-identification. IEEE TCSVT (2025).
> > >
> > > [2] Frequency domain nuances mining for visible-infrared person re-identification. IEEE TIFS (2025).
> > >
> > > [3] Magic tokens: Select diverse tokens for multi-modal object re-identification. CVPR 2024.
> > >
> > > [4] TOP-ReID: Multi-spectral object re-identification with token permutation. AAAI 2024.
> > >
> > > [5] Decoupled feature-based mixture of experts for multi-modal object re-identification. AAAI 2025.
> > >
> > > [6] Mambapro: Multi-modal object re-identification with mamba aggregation and synergistic prompt. AAAI 2025.
> > >
> > > [7] Escaping Modal Interactions: An Efficient DESANet for Multi-Modal Object Re-Identification. IEEE TIP (2025).

---

### Decision · Program_Chairs · 2026-04-30

**Decision:**

Accept (regular)

**Comment:**

This paper has received one Accept and two Weak Accept recommendations. The authors’ rebuttal has addressed the reviewers’ concerns well. I recommend accepting this paper.